# Precision Identification of Locally Advanced Rectal Cancer in Denoised CT Scans Using EfficientNet and Voting System Algorithms

**DOI:** 10.3390/bioengineering11040399

**Published:** 2024-04-19

**Authors:** Chun-Yu Lin, Jacky Chung-Hao Wu, Yen-Ming Kuan, Yi-Chun Liu, Pi-Yi Chang, Jun-Peng Chen, Henry Horng-Shing Lu, Oscar Kuang-Sheng Lee

**Affiliations:** 1Institute of Clinical Medicine, National Yang Ming Chiao Tung University, Taipei 11221, Taiwan; classicpiano2003@gmail.com; 2Division of Colorectal Surgery, Department of Surgery, Taichung Veterans General Hospital, Taichung 40705, Taiwan; 3School of Medicine, National Defense Medical Center, Taipei 11490, Taiwan; 4Institute of Statistics, National Yang Ming Chiao Tung University, Hsinchu 300093, Taiwan; chwu@nycu.edu.tw; 5Institute of Multimedia Engineering, National Yang Ming Chiao Tung University, Hsinchu 300093, Taiwan; pzps0964604.cs10@nycu.edu.tw; 6Department of Post-Baccalaureate Medicine, College of Medicine, National Chung Hsing University, Taichung 402202, Taiwan; b9202080@gmail.com; 7Department of Radiation Oncology, Taichung Veterans General Hospital, Taichung 40705, Taiwan; 8Department of Radiology, Taichung Veterans General Hospital, Taichung 40705, Taiwan; bibichang1023@gmail.com; 9Biostatistics Task Force, Taichung Veterans General Hospital, Taichung 40705, Taiwan; pippan7676@vghtc.gov.tw; 10Department of Statistics and Data Science, Cornell University, Ithaca, NY 14853, USA; 11Stem Cell Research Center, National Yang Ming Chiao Tung University, Taipei 11221, Taiwan; 12Department of Medical Research, Taipei Veterans General Hospital, Taipei 112201, Taiwan; 13Department of Orthopedics, China Medical University Hospital, Taichung 40402, Taiwan; 14Center for Translational Genomics & Regenerative Medicine Research, China Medical University Hospital, Taichung 40402, Taiwan

**Keywords:** rectal cancer, local advanced rectal cancer (LARC), circumferential resection margin (CRM), computed tomography (CT), medical image processing, artificial intelligence (AI)

## Abstract

Background and objective: Local advanced rectal cancer (LARC) poses significant treatment challenges due to its location and high recurrence rates. Accurate early detection is vital for treatment planning. With magnetic resonance imaging (MRI) being resource-intensive, this study explores using artificial intelligence (AI) to interpret computed tomography (CT) scans as an alternative, providing a quicker, more accessible diagnostic tool for LARC. Methods: In this retrospective study, CT images of 1070 T3–4 rectal cancer patients from 2010 to 2022 were analyzed. AI models, trained on 739 cases, were validated using two test sets of 134 and 197 cases. By utilizing techniques such as nonlocal mean filtering, dynamic histogram equalization, and the EfficientNetB0 algorithm, we identified images featuring characteristics of a positive circumferential resection margin (CRM) for the diagnosis of locally advanced rectal cancer (LARC). Importantly, this study employs an innovative approach by using both hard and soft voting systems in the second stage to ascertain the LARC status of cases, thus emphasizing the novelty of the soft voting system for improved case identification accuracy. The local recurrence rates and overall survival of the cases predicted by our model were assessed to underscore its clinical value. Results: The AI model exhibited high accuracy in identifying CRM-positive images, achieving an area under the curve (AUC) of 0.89 in the first test set and 0.86 in the second. In a patient-based analysis, the model reached AUCs of 0.84 and 0.79 using a hard voting system. Employing a soft voting system, the model attained AUCs of 0.93 and 0.88, respectively. Notably, AI-identified LARC cases exhibited a significantly higher five-year local recurrence rate and displayed a trend towards increased mortality across various thresholds. Furthermore, the model’s capability to predict adverse clinical outcomes was superior to those of traditional assessments. Conclusion: AI can precisely identify CRM-positive LARC cases from CT images, signaling an increased local recurrence and mortality rate. Our study presents a swifter and more reliable method for detecting LARC compared to traditional CT or MRI techniques.

## 1. Introduction

Rectal cancer, a significant subset of colorectal malignancies, presents distinct diagnostic and therapeutic challenges, especially in its advanced stages. The treatment approach for rectal cancer is primarily stage-dependent. Stage IV cancer, marked by metastasis to other organs, necessitates systemic therapy. Conversely, Stages I–III are considered localized diseases, and among the treatment modalities, the total mesorectal excision (TME) technique for complete removal of the tumor within the rectal mesentery, introduced by Heald, remains the most effective method to date [1]. According to the research, if a complete excision can be achieved, the likelihood of local recurrence is very low, at 2.6%. A key measure of a complete excision is the pathological absence of cancer cells at the circumferential resection margins (pCRMs). When cancer cells are present at these margins (pCRM+), it is associated with poor prognosis, leading to higher risks of local recurrence and mortality, often due to extensive tumor invasion or inadequate surgical techniques [2]. To reduce the tumor size and facilitate a clean resection, the introduction of radio (chemo)therapy (neoadjuvant therapy) before total mesorectal excision has led to an improved prognosis worldwide [3].

Among these, locally advanced rectal cancer (LARC) is notably challenging due to its deep pelvic location and proximity to critical organ structures. LARC is defined by tumors located within 1 mm of the planned circumferential resection margin (CRM) on imaging, indicating an advanced disease stage. This condition leads to a 26.3% local recurrence rate and a 37.8% mortality rate over a five-year follow-up period, thus complicating treatment efforts [4]. To decrease the local recurrence rate of the disease, applying neoadjuvant chemoradiation therapy before radical resection surgery is suggested in current guidelines. The stronger agent “total neoadjuvant therapy”, which includes standard radiation therapy with six-cycle chemotherapy, is suggested especially for LARC cases [5,6]. However, neoadjuvant treatment increases the expense and the rate of surgical complications, such as an increased rate of anastomosis stricture [7]. The rate of permanent stoma following low anterior resection is 13%. Reducing neoadjuvant therapy to chemotherapy with FOLFOX alone [8] or opting for direct surgery [9] for rectal cancer with an adequate surgical margin has been proven to offer comparable efficacy. Hence, the rapid identification of LARC with margin-threatening characteristics is crucial at the outset of rectal cancer therapy.

Current guidelines recommend magnetic resonance imaging (MRI) for staging rectal cancer; however, the limited availability of MRI resources limits the timeliness of this diagnostic method. Studies have indicated that CT, interpreted by experienced physicians, can achieve comparable efficacy to MRI in assessing the circumferential resection margin (CRM), a crucial factor in LARC diagnosis [10,11]. The rapid advancements in artificial intelligence (AI) image interpretation could significantly speed up and improve diagnostic accuracy for physicians [12,13,14,15,16]. This progress opens a promising avenue for utilizing AI to detect features of LARC that threaten the CRM from CT images.

Currently, the application of AI in cutting-edge methodologies for single-image recognition has shown promising results across numerous studies [17,18]. However, the comprehensive evaluation of case datasets remains an underexplored area in the literature. If AI were capable of identifying cases of locally advanced rectal cancer (LARC) with an accuracy comparable to those of specialists, it could markedly accelerate the treatment initiation process. Moreover, in the context of large-scale, multi-institutional trials for LARC treatment, AI has the potential to minimize diagnostic variations among clinicians, significantly improving the objectivity and fairness of trial outcomes.

Our study explores the use of AI in identifying LARC cases with CRM-threatening features from CT images, aiming to facilitate a rapid diagnosis of LARC. This approach, if successful, could provide a timely and effective tool for the management of rectal cancer in healthcare settings. It could also provide a fair and objective method to define LARC cases in research affairs.

## 2. Materials and Methods

### 2.1. Dataset

We compiled a comprehensive dataset of rectal surgeries conducted from 1 October 2010 to 31 December 2022. We focused on first-diagnosed rectal cancer cases in clinical stages T3–4, prioritizing CT images of high quality and with contrast. Exclusions included cases with primary tumors in other organs, non-adenocarcinoma pathology, tumors located in the sigmoid colon or lower rectum, early-stage T0–2 tumors, and CT images lacking contrast or compromised by prosthesis noise. The CT images around the rectal tumor segment were harvested. This selection yielded data from 1070 patients, providing 7739 CT images. These images were categorized into three sets: 739 cases for training, 134 for test set 1, and 197 for test set 2 consisting of stage-2–3 patients who underwent direct surgery (Figure 1, Table 1). Ethical approval was obtained (IRB number: CE21235B), and this study was registered at ClinicalTrials.gov (NCT05723965) accessed on 9 February 2023. 

### 2.2. CT Imaging Methods

CT scans were conducted at diagnosis and pre-surgery using Philips Healthcare scanners (Brilliance series, Cleveland, USA) at 100, 120, and 130 kV, or with automatic mA control, and without additional noise reduction. Images were captured at a slice thickness of 0.7 to 1.0 mm and a resolution of 512 × 512 pixels. The contrast medium dosage was calculated based on body weight, with a maximum limit of 100 mL. All images were reconstructed into 5 mm slices for detailed interpretation and analysis.

### 2.3. Annotation of CRM-Threatening LARC Cases

We defined LARC as tumor invasion within 1 mm of the circumferential resection margin (CRM) in MRI or CT images (Figure 2a,b). To validate the reliability of CT, fifty rectal cancer patients underwent both CT and MRI, and three specialists (CY Lin, 7 years of proctology; YC Liu, 8 years of radio-oncology; PY Chang, 12 years of radiology) assessed the LARC status. The consistency between CT and MRI interpretations was assessed using a kappa statistic of 0.728 (IBM SPSS version 22.0), indicating substantial agreement.

### 2.4. Image Processing

We used the data collected above to train the deep learning algorithm model. The training materials included the original DICOM files downloaded from the PACS system. We utilized 3D slicer (version 5.2.2) software for image cropping along the outer pelvic edge and inner pelvic edge. After training and testing, it was found that using only the images cut along the inner pelvic edge yielded the highest result, based on preliminary AUC results (Figure 2c, Table 2).

First, we normalized the CT images. Since the CT observation range was too large, we limited its range by normalizing the image with a window level of 50 and a window width of 400. A value of 250 and a minimum value of −150 were substituted, which are the standard observation values for the abdomen. Although these values may vary by institution and provider, the window width and level are usually very similar. After normalization, we scaled the values to a range of 0–255 to match the storage range of grayscale images. In addition to the normalization of CT values, we also normalized the unit distance of CT images. Different CT images may have different unit distances due to different operators, so we corrected the unit distance of all images to 1 mm × 1 mm per pixel, ensuring that all images had the same unit size. Improving image quality, specifically through noise reduction, is a crucial step prior to performing CT image prediction. From the literature [19,20,21,22], it is confirmed that the noise in CT images is typically additive white Gaussian noise. We selected nonlocal mean filtering for denoising, as it searches for similar areas in the image in units of image blocks and then averages these areas, which can better filter out the Gaussian noise in the image. We set the search area to 21 × 21 pixels and the similarity comparison block to 7 × 7 pixels, and we used the square of the pixel brightness difference to estimate the similarity (Figure 3).

After removing the noise from the image, we then applied dynamic histogram equalization to improve its quality [23]. It first applies a smoothing process to the histogram using a 1 × 3 filter, and then searches for the locations of the troughs in the histogram to perform the first histogram division. To ensure that there were no dominant pixel values within each division (i.e., to prevent high-frequency occurrences of luminance values from overriding low-frequency occurrences), an evaluation of the segments was performed, followed by a second histogram division. During the second division, the mean and standard deviation of each segment were calculated. If the standard deviation was more than twice the 68.3% probability threshold, no further division was performed. If the condition was not met, the segment was divided into three subsegments.

Finally, histogram equalization was applied. This processing method solves the problem of high-frequency luminance values overpowering low-frequency luminance values during histogram equalization, thereby achieving a superior contrast enhancement effect. The aim is to locate the tumor in the resulting image after final processing. Since the image is cropped around the tumor, we assume that the contour closest to the center is the contour of the tumor. Therefore, we started searching for contours from the center of the image outward. We used the OpenCV package to find all the contours by checking the distance from the contour’s centroid to the center of the image. We continuously updated and iterated until we found the closest contour. We drew the contour on the original image and used it as one of the inputs to our model [24].

### 2.5. Deep Learning Algorithm for CRM-Positive Image Identification

This study employed the deep learning architecture EfficientNetB0, recognized for its efficiency and high performance despite a manageable parameter size, making it ideal for fine-tuning with the limited yet high-quality data collected in the hospital setting. We utilized EfficientNetB0 as a pre-trained model, integrating a global average pooling layer immediately preceding the classification layer, and replacing the traditional fully connected layer [25]. The loss was set to binary cross-entropy, and the class weights were set to adjust for data imbalance. We used Adam as the optimizer, which updates parameters of different scales based on different gradients. We set the learning rate to 0.001 and the batch size to 32. We trained the model for 500 epochs and selected the model with the best validation accuracy. We evaluated the impact of the architectural decisions on the overall performance of the pipeline by monitoring the performance in the training set. The impact of the hyperparameter values on the generalization capabilities of the models in an ablation study was investigated in the performance monitoring in the validation and test sets. We divided the training set data into 64%, 16%, and 20% for training, validation, and testing, respectively, based on the proportion of patients. Since the amount of data provided by each patient was different, there may be errors in the proportion of categories after splitting. We used a hash table to record the number of positive and negative images corresponding to each patient and recursively divided the data until the proportion of each category was close to the minimum error (Figure 3).

### 2.6. Determining LARC Cases through Series of Images

We used a voting system to gather statistics and determine the presence of LARC based on the proportion of CRM-positive images in each case. The determination of whether a case was considered locally advanced rectal cancer (LARC) was based on the collective clinical judgment of three specialists. Hard voting was used at first; that is, if the probability of the model judging the image as positive was greater than or equal to 0.5, the image was considered positive, and whether the patient was LARC was determined according to the previous threshold.

Hard voting

Any one positive: any CRM-positive image be predicted in case series:$$0.20\;(one\text{-}fifth):\frac{\Sigma (Number\;of\;CRM\;positive\;image\;be\;predicted)}{number\;of\;images\;(n)}>1/5$$
$$0.25\;(one\text{-}fourth):\frac{\Sigma (Number\;of\;CRM\;positive\;image\;be\;predicted)}{number\;of\;images\;(n)}>1/4$$
$$0.33\;(one\text{-}third):\frac{\Sigma (Number\;of\;CRM\;positive\;image\;be\;predicted)}{number\;of\;images\;(n)}>1/3$$
$$0.50\;(one\text{-}half):\frac{\Sigma (Number\;of\;CRM\;positive\;image\;be\;predicted)}{number\;of\;images\;(n)}>1/2$$

Subsequently, soft voting was adopted, averaging the probability values across all images of a patient. The optimal threshold was determined using the Youden index on ROC curves, combining sensitivity and specificity to assess the diagnostic effectiveness.

Soft voting
$$Any\;one\;positive:\frac{\Sigma (Predicted\;risk\;of\;CRM\;positive)}{number\;of\;images\;(n)}>1/n$$
$$0.20\;(one\text{-}fifth):\frac{\Sigma (Predicted\;risk\;of\;CRM\;positive)}{number\;of\;images\;(n)}>1/5$$
$$0.25\;(one\text{-}fourth):\frac{\Sigma (Predicted\;risk\;of\;CRM\;positive)}{number\;of\;images\;(n)}>1/4$$
$$0.33\;(one\text{-}third):\frac{\Sigma (Predicted\;risk\;of\;CRM\;positive)}{number\;of\;images\;(n)}>1/3$$
$$0.50\;(one\text{-}half):\frac{\Sigma (Predicted\;risk\;of\;CRM\;positive)}{number\;of\;images\;(n)}>1/2$$

### 2.7. Local Recurrence Rate Analysis

Training AI models to expert-level judgment is not sufficient in demonstrating clinical value; we aimed to understand the clinical predictive power of the decisions made. In rectal cancer, LARC has significantly higher local recurrence (LR) and mortality rates. Therefore, we selected 197 stage-2–3 rectal cancer patients who underwent direct surgery (whose survival rates are not influenced by stage or treatment) from test set 2, followed for an average of 49.3 months (mean). We conducted survival analysis based on the AI model’s prediction to compare the predictive strength of AI and specialist physicians’ interpretations.

### 2.8. Statistical Analysis

Data were analyzed from 1 May 2022 to 31 May 2023, using Cohen’s kappa statistics (SPSS version 22.0) for CT and MRI inter-rater reliability, chi-square tests, and ANOVA for patient characteristic comparisons. Survival curves were evaluated using the Kaplan–Meier method and log-rank tests, employing IBM SPSS version 22.0.

This methodology ensured a robust and comprehensive approach to evaluating AI’s capability in accurately identifying LARC cases through CT imaging, potentially transforming diagnostic processes in oncological care.

## 3. Results

### 3.1. Training Set and Test Set Materials

In our training set, we compiled 739 cases, including 347 (47.0%) identified as LARC. From these cases, we obtained 5464 CT images, with 1897 (36.7%) classified as CRM-positive and 3267 (63.3%) as CRM-negative. Using the cropping method focused on the inner pelvic edge, we processed these images for training using the ResNet50 model, a multiscale squeeze and excitation model. For test set 1, we gathered 134 cases with 63 (47.0%) LARC instances, resulting in 1307 CT images (481 CRM-positive and 826 CRM-negative). Test set 2 comprised 197 cases, including 108 (54.8%) LARC cases, yielding 1268 images (502 CRM-positive, 766 CRM-negative) (Table 1).

### 3.2. Model Performance by Image

Upon completing image processing, we fed the data into our deep learning algorithm. In test set 1, the AI model achieved a sensitivity, specificity, accuracy, and balanced accuracy of 0.81, with an AUC of 0.89. For test set 2, the model achieved a sensitivity of 0.75, specificity of 0.81, accuracy of 0.79, balanced accuracy of 0.78, and AUC of 0.86. These results illustrate the AI’s capability in accurately interpreting CRM-positive CT images, closely aligning with the performance of experienced specialists (Table 3, Appendix A).

### 3.3. Model Performance by Patient

We next assessed whether the AI’s ability to interpret LARC cases paralleled that of specialists. A voting system was employed for this purpose, initially using hard voting (binary classification of images). The optimal AI performance in test set 1 was achieved with a one-fifth threshold, resulting in an AUC of 0.84 and a binary accuracy (BA) of 0.84. In test set 2, the best performance was at a one-fourth threshold, with an AUC of 0.79 and a BA of 0.79. Switching to soft voting, which uses the summation of probabilities for assessments, improved the AI’s performance: the highest performance was at a one-third threshold, achieving an AUC of 0.93 and BA of 0.88 in test set 1 and an AUC of 0.88 and BA of 0.83 in test set 2 (Table 4, Appendix A).

### 3.4. Expanding the Training Set

To enhance the model’s capabilities, we combined the data from testing sets 1 and 2 to form new training sets for Models 2a and 2b. Model 2a was trained with 936 patients (6732 images, 123.2% more than the original training set) and Model 2b with 873 patients (6771 images, 123.9% more than the original training set). For Model 2a in test set 1, we noted an image identification BA of 0.83 and an AUC of 0.89. The best LARC identification was achieved at a one-third threshold in hard voting (BA: 0.85, AUC: 0.86) and a half threshold in soft voting (BA: 0.89, AUC: 0.94). In test set 2, Model 2b’s image identification showed a BA of 0.77 and AUC of 0.85. The optimal LARC identification occurred at a one-fourth threshold in hard voting (BA: 0.80, AUC: 0.80) and at one-third and one-fifth thresholds in soft voting (BA: 0.82/0.83, AUC: 0.88) (Figure 4, Table 5, Appendix A).

### 3.5. Prediction Results and Survival Analysis in Test Set 2

Regarding patients with stage 2 or 3 rectal cancer who underwent surgical treatment, over an average follow-up period of 49.3 months, it was observed that locally advanced rectal cancer (LARC), as interpreted by physicians, was associated with higher rates of local recurrence (LR) and mortality, though these findings were not statistically significant (*p* = 0.106; 0.172).

When LARC was identified using artificial intelligence (AI), the number of LARC cases identified decreased as the risk threshold was increased (from 108 cases at a threshold of 0.20 to 51 cases at a threshold of 0.50). Despite the reduction in identified cases, there was a consistent trend of higher LR rates among the identified LARC cases across all risk thresholds. Notably, the trend of increased LR risk became significantly more pronounced when the threshold was raised from 0.20 to 0.50 (*p* = 0.052 to *p* = 0.003). The predictive value for mortality, when compared with physician interpretation, showed a similar trend towards higher rates but was not statistically significant (Figure 5, Table 6, Appendix A). This result demonstrates that within the threshold range of 0.20 to 0.50, AI’s predictions consistently showed clinical significance, indicating that AI can reliably interpret meaningful outcomes.

### 3.6. Visual Examples of Interpretation by AI and Doctor

Figure 6 presents a series of image cases with interpretations from both a doctor and artificial intelligence (AI) across various thresholds. The evaluated clinical outcomes include the positive pathological circumferential margin (pCRM), the time until local recurrence (LR), and overall survival (OS).

For Case 359, interpretations by both the doctor and AI indicated positive results, leading to positive pCRM and local recurrence at 7.9 months, respectively, with overall survival times of 39.5 months. Case 300 highlights a scenario where the AI identified a few images with a positive CRM, which was overlooked by the doctor. This patient experienced a local recurrence at 12.3 months and succumbed to the disease at 34.0 months.

In Case 458, which featured a higher-positioned rectal cancer near the small intestine, the physician determined that surgical resection would not pose a risk of positive circumferential resection margin (pCRM), suggesting confidence in achieving clear margins and thus reducing the likelihood of local recurrence. Despite this professional assessment, an artificial intelligence (AI) system identified the cancer as locally advanced rectal cancer (LARC) at risk thresholds ranging from 0.20 to 0.33. This classification by the AI was likely influenced by the visual proximity of the rectum to the small intestine in several images, which could be interpreted as a more aggressive or advanced disease. However, following surgery, the patient was found to have stage II cancer and did not experience any disease recurrence over a prolonged period.

## 4. Discussion

### 4.1. Integration of Key Results with Existing Research

Our study’s crucial discovery is that AI can accurately interpret CT images for the diagnosis of locally advanced rectal cancer (LARC), despite the absence of a comparative model. By conducting a comparative analysis with the results from the MERCURY trial(4), which used MRI to assess rectal cancer, we found that local recurrence rates for LARC were comparable between our study and the MERCURY trial (22.9% vs. 26.3%). This comparison emphasizes the AI’s predictive accuracy, showcasing its potential value even without direct model comparisons. Moreover, our AI model’s high sensitivity, specificity, accuracy, and predictive value in differentiating CRM-positive from CRM-negative images are comparable to the performance of experienced radiologists. This underscores the substantial promise of AI in enhancing medical diagnostics.

### 4.2. Significant Achievements and Contributions

This research makes a substantial contribution by applying AI in series analysis of medical images, a domain traditionally reliant on human expertise. The use of hard and soft voting techniques enables the AI model to interpret multiple images for a comprehensive diagnosis, highlighting its potential in complex diagnostic scenarios. This extends AI’s application beyond state-of-the-art methods [26,27] such as single-image analysis, demonstrating more intricate diagnostic tasks.

Moreover, this study’s findings on the AI model’s ability to predict local recurrence risks in LARC cases underscore the potential role of AI in clinical decision making and prognostication, aligning with recent advances in AI for personalized treatment planning [28,29,30,31,32].

### 4.3. Combining Current Findings with Original Study Aspects

For newly diagnosed rectal cancer, understanding whether there is organ metastasis and evaluating local staging are crucial. LARC, defined as CRM-threatening, necessitates neoadjuvant chemoradiation therapy [3,5]. While MRI is the recommended imaging modality [4,10], its limited availability has necessitated reliance on CT scans. Our study’s use of CT interpretations by experienced physicians for deep learning algorithm training shows that AI can quickly detect LARC with enough data, providing a viable alternative to MRI.

Identifying LARC involves understanding the spatial relationship between the cancer and other pelvic structures. Our approach, using 2D image feature recognition and voting systems, successfully tackles this challenge, offering a method that closely matches the accuracy of professional physicians’ interpretations.

### 4.4. State-of-the-Art Method for CRM+ Images and LARC Cases

Our study introduces an adaptation of the EfficientNetB0 architecture and a novel voting mechanism for detecting locally advanced rectal cancer (LARC) via CT scans. EfficientNetB0 is engineered to maximize accuracy with minimal parameters, tackling the challenge of scarce high-quality medical imaging data and representing state-of-the-art methods in recognizing CRM-positive images. To further enhance LARC case identification, we employed our innovative voting system, combining hard and soft voting approaches. Typically, a case consists of 3 to 10 images, and the challenge lies in determining LARC from a series of image results. Accurately interpreting CRM+ images does not automatically equate to identifying an LARC case, a finding corroborated by our study results; indeed, any single prediction result is often the least accurate. We discovered that employing a voting system for comprehensive risk assessment yields consistent and precise predictive outcomes. This system closely mirrors expert decision making by comprehensively evaluating image series, thereby providing consistent results across different thresholds and ensuring reliable LARC identification. Given the absence of comparative methods for LARC case identification, our CRM+ identification approach with the soft voting system is considered a state-of-the-art method. Importantly, this study goes beyond mere diagnostic accuracy to demonstrate the model’s clinical utility in predicting local recurrence rates and overall survival, bridging the gap between technical accuracy and meaningful clinical outcomes. The strategic data management and model training approach further ensure the system’s operational efficiency and reliability. This integration of innovative architecture and practical clinical relevance marks a significant step forward in leveraging AI for medical diagnostics.

### 4.5. How to Utilize the AI Prediction Results

In the last three example cases, there was a significant discrepancy between the AI’s predictive assessment and the actual clinical course. This underscores the critical need for the careful evaluation of AI recommendations, especially when they might lead to the overestimation of disease severity based on imaging alone. While AI can provide valuable insights, particularly in complex cases, this scenario highlights its limitations and the irreplaceable value of human clinical judgment in making final treatment decisions. These case outcomes exemplify the importance of integrating AI tools with comprehensive clinical evaluation to ensure accurate diagnosis and appropriate treatment planning.

### 4.6. Limitations and Future Directions

This study utilized CT images from a single institution, and the limited number of cases could potentially restrict the robustness of its findings. A larger, well-annotated database could enhance the outcomes, offering a richer dataset for analysis and potentially improving the accuracy of AI predictions. Furthermore, this research could be expanded to include MRI imaging in the future. With precise annotation, the superior detail offered by MRI images could lead to the more accurate identification of locally advanced rectal cancer (LARC). Moreover, MRI’s enhanced imaging capabilities could also contribute to the investigation of lymph node involvement, significantly aiding in disease staging.

### 4.7. Possible Applications of this Research

Utilizing this approach, it may be possible to develop a real-time monitoring system that alerts physicians to the presence of locally advanced rectal cancer (LARC) while they are reviewing CT images. This system could offer objective and immediate feedback, serving as an invaluable tool in clinical settings. Furthermore, during multicenter studies, it could act as an objective reference standard for diagnosing LARC.

Expanding on this foundation, future research that integrates semi-automated segmentation tools [33], localizes colorectal cancer [34], and detects colon polyps [35] in contrast enhancement CT could lead to the creation of automated imaging staging for colorectal cancer. AI has the potential to provide an experienced and objective perspective, serving as a valuable tool in clinical treatment. Such a system could enhance clinical treatment by providing consistent and accurate staging information. In the future, with the aid of GAN [36], it will be possible to more accurately simulate the relationship between rectal tumors and nearby organs, thus defining LARC. By automating the staging process and integrating real-time risk assessment, clinicians can improve the precision of their diagnoses and treatment strategies, leading to better patient outcomes. This illustrates the promise of combining AI with medical expertise to advance healthcare delivery.

## 5. Conclusions

This study highlights the potential of AI in accurately interpreting CT images for diagnosing locally advanced rectal cancer (LARC), rivaling the precision of experienced radiologists. It underscores the need for further automation and broader datasets in AI diagnostics, particularly in settings where MRI is less accessible. Ultimately, these findings pave the way for integrating AI into routine oncological care, enhancing diagnostic efficiency and accuracy.

## Figures and Tables

**Figure 1 bioengineering-11-00399-f001:**
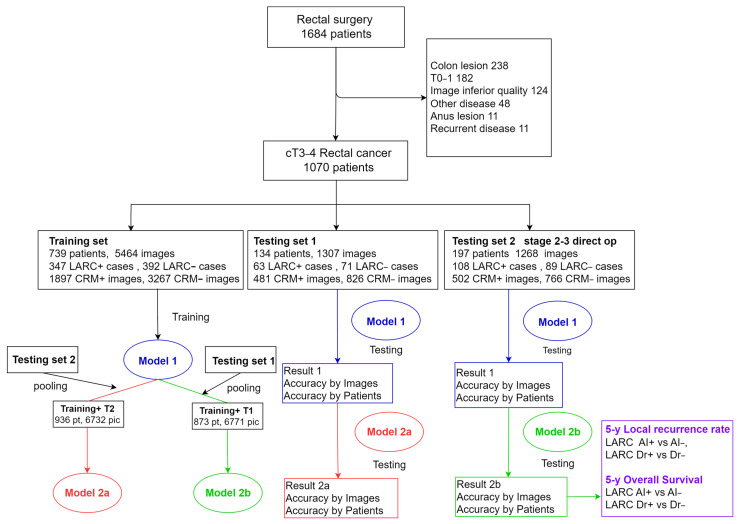
Diagrammatic representation of study material workflow. This figure outlines the systematic process used in this study, tracing the flow of materials from initial data collection through to their final application within the research framework.

**Figure 2 bioengineering-11-00399-f002:**
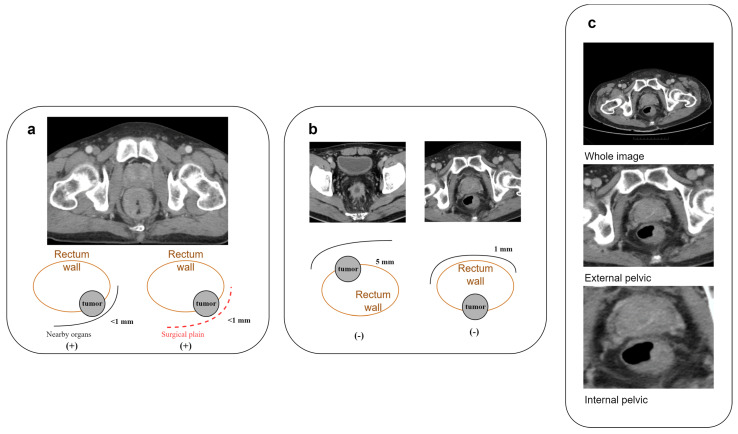
Characterization and processing of rectal cancer features in CT imaging. (**a**) Criteria for identifying CRM-threatening features associated with locally advanced rectal cancer in CT scans; (**b**) illustrative cases of rectal cancer in CT imagery lacking CRM-threatening attributes; (**c**) comparative illustration of three cropping techniques for rectal cancer image preparation: full image, external pelvic boundary, and internal pelvic boundary.

**Figure 3 bioengineering-11-00399-f003:**
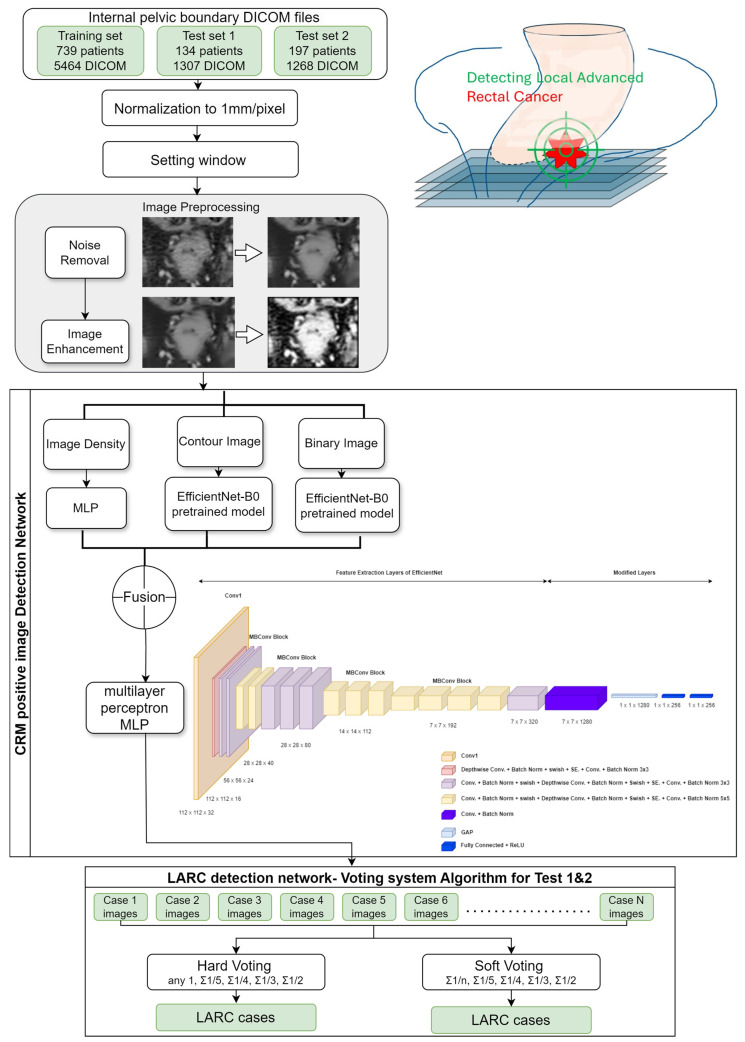
AI-based diagnostic framework for detecting the locally advanced rectal cancer architectural blueprint of the AI model for LARC identification. This diagram illustrates the methodological approach and the deep learning architecture utilized in this study for the detection of locally advanced rectal cancer (LARC) from CT scans. The framework encapsulates the AI’s training and predictive process, detailing the progression from image preprocessing to the final LARC classification.

**Figure 4 bioengineering-11-00399-f004:**
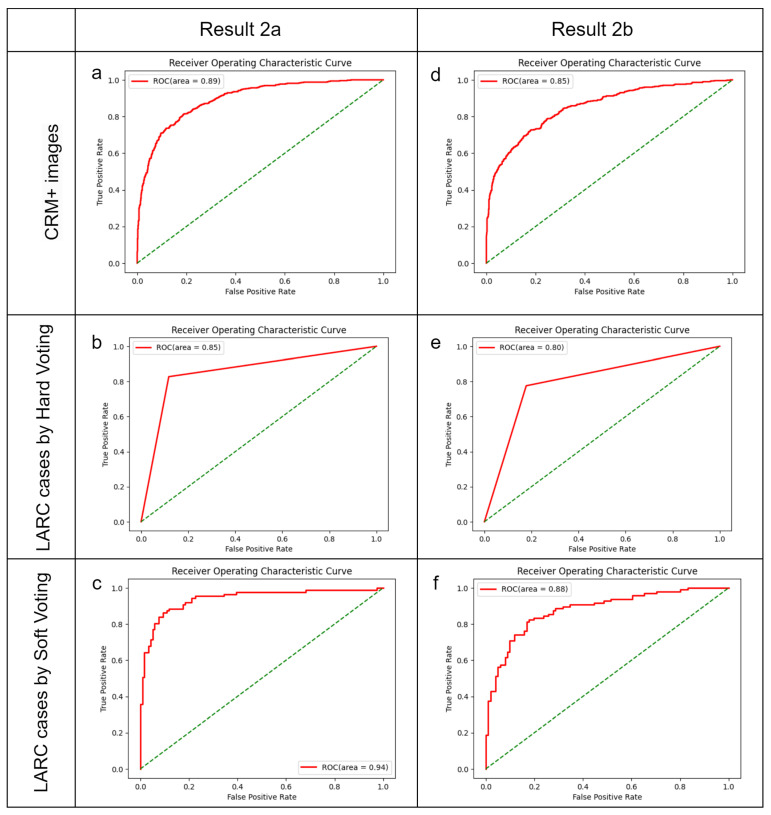
Optimal diagnostic outcomes using Models 2a and 2b. The green dotted line indicate randomize classifier. (**a**–**c**) Performance of Model 2a, amalgamating data from the training set and test set 2, when applied to test set 1. (**a**) Identification of CRM-threatening features indicative of LARC; (**b**) determination of LARC status using a hard voting threshold of one-third; (**c**) assessment of LARC cases employing a soft voting threshold of a half. (**d**–**f**) Efficacy of Model 2b, integrating data from the training set and test set 1, utilized on test set 2. (**d**) Image analysis for CRM-threatening features associated with LARC; (**e**) LARC case adjudication based on a hard voting threshold of one-fourth; (**f**) LARC case determination via a soft voting threshold of one-third.

**Figure 5 bioengineering-11-00399-f005:**
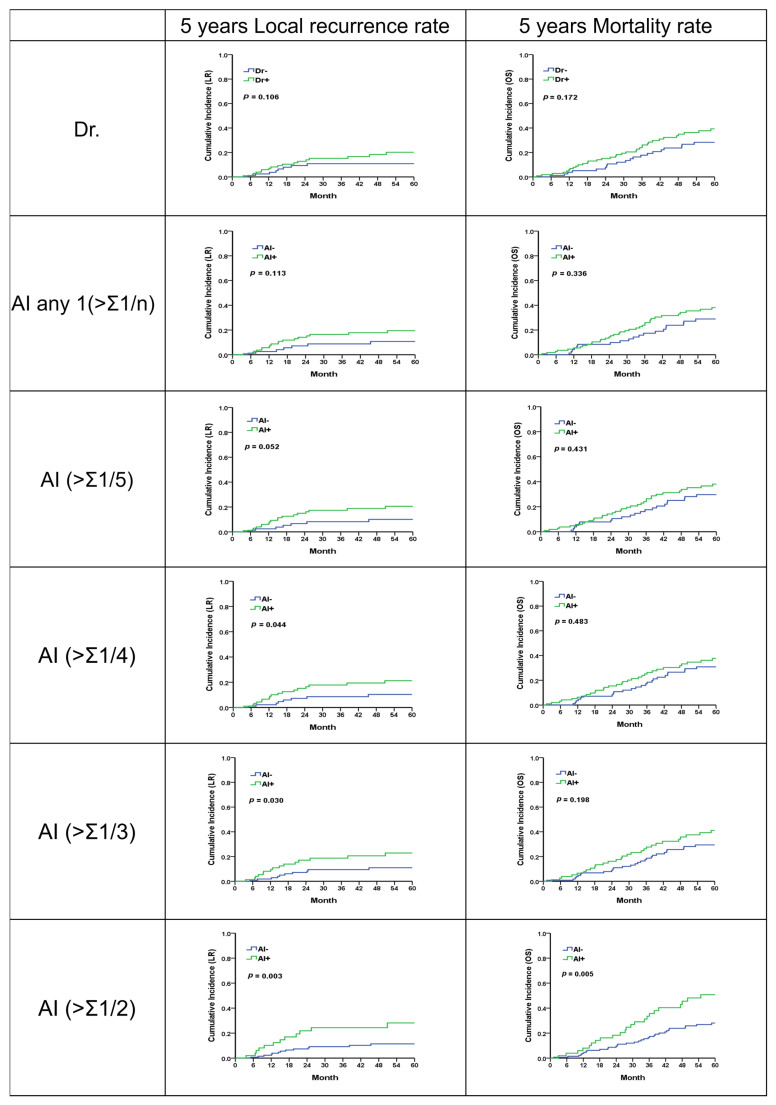
Five-year local recurrence and overall survival rates for LARC from physician and analysis of different AI methodologies.

**Figure 6 bioengineering-11-00399-f006:**
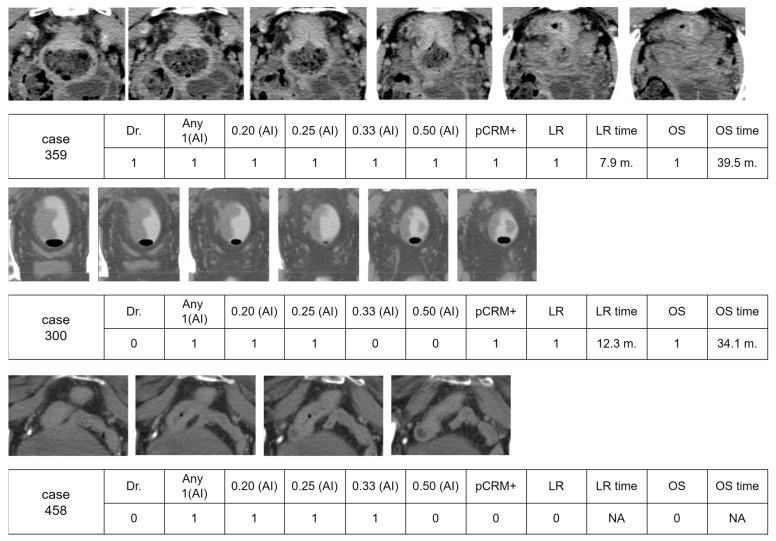
Three visual examples of interpretation results by a physician and AI. The surgery outcome with a positive pathological circumferential margin (pCRM), disease survival of local recurrence, and overall survival time were also recorded. The following abbreviations are used: pCRM+, positive pathological circumferential margin; LR, local recurrence; OS, overall survival.

**Table 1 bioengineering-11-00399-t001:** Demographic, clinical, and pathological characteristics of patients in study group.

		Training (n = 739)		Testing 1 (n = 134)		Testing 2 (n = 197)		*p* Value
**Gender**	Female	283	(38.3%)	53	(39.6%)	73	(37.1%)	0.898
	Male	456	(61.7%)	81	(60.4%)	124	(62.9%)	
**Age**		66.40	±14.02	68.88	±12.18	69.86	±13.92	0.003 **
**Site of lesions**	Upper	169	(22.9%)	45	(33.6%)	70	(35.5%)	<0.001 **
	Middle	257	(34.8%)	49	(36.6%)	81	(41.1%)	
	Lower	313	(42.4%)	40	(29.9%)	46	(23.4%)	
**Clinical T stage**	T3	554	(79.0%)	98	(74.8%)	170	(86.3%)	0.024 *
	T4	147	(21.0%)	33	(25.2%)	27	(13.7%)	
**Clinical N stage**	N0	197	(26.7%)	33	(24.6%)	108	(54.8%)	<0.001 **
	N1–2	542	(73.3%)	101	(75.4%)	89	(45.2%)	
**Clinical Stage**	2	179	(24.2%)	31	(23.1%)	108	(54.8%)	<0.001 **
	3	454	(61.4%)	65	(48.5%)	89	(45.2%)	
	4	106	(14.3%)	38	(28.4%)	0	(0.0%)	
**c CRM**	negative	392	(53.0%)	71	(53.0%)	89	(45.2%)	0.936
	positive	347	(47.0%)	63	(47.0%)	108	(54.8%)	
**c CRM+ image**	negative	3267	(63.3%)	826	(63.2%)	766	(60.4%)	
	positive	1897	(36.7%)	481	(36.8%)	502	(39.6%)	
**Radiation therapy**		279	(37.8%)	36	(26.9%)	0	(0.0%)	<0.001 **
**Operation**		683	(92.4%)	134	(100.0%)	197	(100.0%)	<0.001 **
**Pathology T**	T0–1	11	(1.6%)	0	(0.0%)	5	(2.5%)	<0.001 **
	T2	30	(4.4%)	5	(3.7%)	46	(23.4%)	
	T3	512	(75.0%)	96	(71.6%)	122	(61.9%)	
	T4	130	(19.0%)	33	(24.6%)	24	(12.2%)	
**Pathology N**	N0	175	(25.6%)	33	(24.6%)	85	(43.4%)	<0.001 **
	N1–2	508	(74.4%)	101	(75.4%)	111	(56.6%)	
**Pathology Stage**	1	0	(0.0%)	0	(0.0%)	32	(16.2%)	<0.001 **
	2	161	(23.6%)	30	(22.4%)	53	(26.9%)	
	3	423	(61.9%)	66	(49.3%)	112	(56.9%)	
	4	99	(14.5%)	38	(28.4%)	0	(0.0%)	
**p CRM**	negative	574	(84.0%)	98	(73.1%)	172	(87.8%)	0.001 **
	positive	109	(16.0%)	36	(26.9%)	24	(12.2%)	

Chi-square test or ANOVA test. * *p* < 0.05, ** *p* < 0.01.

**Table 2 bioengineering-11-00399-t002:** Prediction rates of CRM-positive images in test set 1 derived from various image material sources.

	Sensitivity	Specificity	Accuracy	AUC
whole picture	0.44	0.702	0.621	0.59
external pelvis	0.48	0.784	0.67	0.7
internal pelvis	0.527	0.816	0.713	0.77
image processed	0.811	0.809	0.81	0.89

**Table 3 bioengineering-11-00399-t003:** Prediction of CRM-positive images by Model 1.

Image-Based	Sensitivity	Specificity	Accuracy	Balanced Accuracy	AUC
Training and validation (739 series, 5164 images)	0.80	0.82	0.82	0.81	0.9
Testing 1 (134 series, 1307 images)	0.81	0.81	0.81	0.81	0.89
Testing 2 (197 series, 1268 images)	0.75	0.81	0.79	0.78	0.86

**Table 4 bioengineering-11-00399-t004:** Prediction of LARC cases using Model 1 with hard and soft voting systems.

Testing 1 Set	Threshold	Sensitivity	Specificity	Accuracy	Balanced Accuracy	AUC
	any 1	0.96	0.65	0.80	0.81	0.81
	>Σ1/5	0.90	0.78	0.84	0.84	0.84
hard voting	>Σ1/4	0.85	0.83	0.84	0.84	0.84
	>Σ1/3	0.76	0.89	0.83	0.82	0.82
	>Σ1/2	0.65	0.91	0.79	0.78	0.78
	any 1 (>Σ1/n)	0.88	0.74	0.81	0.81	0.89
	>Σ1/5	0.91	0.78	0.84	0.85	0.91
soft voting	>Σ1/4	0.85	0.82	0.84	0.84	0.91
	>Σ1/3	0.87	0.89	0.88	0.88	0.93
	>Σ1/2	0.76	0.91	0.86	0.83	0.93
**Testing 2 Set**						
	Any 1	0.90	0.61	0.77	0.75	0.75
	>Σ1/5	0.85	0.72	0.79	0.79	0.79
hard voting	>Σ1/4	0.82	0.76	0.80	0.79	0.79
	>Σ1/3	0.73	0.85	0.79	0.79	0.79
	>Σ1/2	0.57	0.91	0.73	0.74	0.74
	any 1 (>Σ1/n)	0.84	0.70	0.78	0.77	0.86
	>Σ1/5	0.85	0.74	0.80	0.80	0.88
soft voting	>Σ1/4	0.81	0.77	0.79	0.79	0.87
	>Σ1/3	0.78	0.87	0.83	0.83	0.88
	>Σ1/2	0.67	0.90	0.81	0.78	0.87

**Table 5 bioengineering-11-00399-t005:** Identification of CRM-positive images and LARC cases by Model 2a (trained with training set + test set 2) and Model 2b (trained with training set + test set 1).

Result 2a of Testing Set 1		Sensitivity	Specificity	Accuracy	Balanced Accuracy	AUC
Image-based		0.76	0.87	0.82	0.83	0.89
	any 1	0.94	0.66	0.81	0.80	0.80
	>Σ1/5	0.90	0.79	0.84	0.84	0.84
hard voting	>Σ1/4	0.87	0.83	0.85	0.85	0.85
	>Σ1/3	0.83	0.88	0.86	0.85	0.86
	>Σ1/2	0.70	0.92	0.84	0.81	0.81
	any 1 (>Σ1/n)	0.84	0.85	0.84	0.84	0.91
	>Σ1/5	0.90	0.82	0.86	0.86	0.93
soft voting	>Σ1/4	0.81	0.92	0.87	0.86	0.93
	>Σ1/3	0.86	0.91	0.89	0.88	0.94
	>Σ1/2	0.92	0.86	0.88	0.89	0.94
**Result 2b of Testing Set 2**		**Sensitivity**	**Specificity**	**Accuracy**	**Balanced Accuracy**	**AUC**
Image-based		0.68	0.86	0.78	0.77	0.85
	any 1	0.88	0.69	0.80	0.78	0.78
	>Σ1/5	0.82	0.76	0.80	0.79	0.79
hard voting	>Σ1/4	0.78	0.82	0.80	0.80	0.80
	>Σ1/3	0.55	0.93	0.75	0.74	0.74
	>Σ1/2	0.64	0.90	0.80	0.77	0.77
	any 1 (>Σ1/n)	0.78	0.86	0.81	0.82	0.87
	>Σ1/5	0.81	0.85	0.83	0.83	0.88
soft voting	>Σ1/4	0.80	0.84	0.82	0.82	0.87
	>Σ1/3	0.82	0.82	0.82	0.82	0.88
	>Σ1/2	0.85	0.73	0.78	0.79	0.86

**Table 6 bioengineering-11-00399-t006:** Summary of disease survival time in interpretation of physician and AI in each threshold using soft voting. The disease survival status was determined using the local recurrent time and overall survival time.

Model		Total	Local Recurrence (n)	Censored		LR Rate (%)			*p*	Overall Survival (Mortality)	Censored		Overall Survival (Mortality) Rate			*p*
				(n)	%	1y	3y	5y			(n)	%	1y	3y	5y	
Dr	−	89	8	81	91.0	97.5	89.2	89.2	0.106	24	65	73.0	96.2	82.2	71.7	0.172
	+	108	18	90	83.3	94.0	84.8	79.8		41	67	62.0	94.1	76.2	60.7	
AI any 1 (>Σ1/n)	−	82	7	75	91.5	97.4	91.2	89.2	0.113	23	59	72.0	95.8	82.6	71.1	0.336
	+	115	19	96	83.5	94.3	83.6	80.5		42	73	63.5	94.5	76.2	61.8	
AI (>Σ1/5)	−	89	7	82	92.1	97.6	91.8	90.0	0.052	26	63	70.8	96.1	82.4	70.3	0.431
	+	108	19	89	82.4	94.0	82.7	79.5		39	69	63.9	94.2	75.9	61.9	
AI (>Σ1/4)	−	99	8	91	91.9	97.8	91.4	89.6	0.044	29	70	70.7	96.5	82.8	69.1	0.483
	+	98	18	80	81.6	93.4	82.2	78.8		36	62	63.3	93.7	74.9	62.2	
AI (>Σ1/3)	−	118	10	108	91.5	98.1	90.6	89.1	0.030	32	86	72.9	96.2	82.5	70.6	0.198
	+	79	16	63	79.7	91.9	81.3	77.1		33	46	58.2	93.5	73.8	58.9	
AI (>Σ1/2)	−	146	13	133	91.1	97.7	90.8	88.6	0.003	39	107	73.3	96.2	83.6	72.0	0.005
	+	51	13	38	74.5	89.8	75.6	71.9		26	25	49.0	92.0	66.5	49.3	

## Data Availability

The research data supporting the findings of this publication are available upon reasonable request. Interested parties may contact either the corresponding author or the first author for access to these materials.

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
