# Peer review of "Precision Identification of Locally Advanced Rectal Cancer in Denoised CT Scans Using EfficientNet and Voting System Algorithms"

_bioengineering, 2024, doi:10.3390/bioengineering11040399_

Round 1

Reviewer 1 Report

Comments and Suggestions for Authors   The proposed study presents innovative research aspects, but the authors must focus better on the goals and motivations of the research in the introductory section. In particular, they must describe in more detail what the performance advantages are compared to related models in the literature in terms of accuracy and precision in the detection of CRM-threatening features from CT images.     The framework schematized in FIg. 3 must be described in detail. In particular, you must describe all included functional components.   How were training and test sets extracted? Was a random partitioning of the dataset carried out or were they selected in order to contain the differently classified images in comparable percentages? A discussion is needed on this point.   By analyzing the prediction results shown in Fig. 5 it is not possible to highlight the variations in the trends of the local recurrent rate and the survival rate obtained using the three asking methods. It is necessary to discuss, also with the help of a summary table, how these trends differ for the three methods. Comments on the Quality of English Language

English typos in the text must be corrected.

Author Response

Dear Reviewer

Thank you for your valuable feedback on our manuscript. We have carefully considered your comments and have made the following revisions to address your concerns:

  1. Introduction - Goals and Motivations: We have revised the introductory section to provide a clearer statement of the goals and motivations of our research. Our intention is to articulate the significance of this study in advancing the field and to outline the novel aspects of our work. This revision is aimed at ensuring readers understand the context and importance of our research contributions.
  2. Comparison with Previous Studies: We acknowledge your concern regarding the absence of a direct model comparison due to the pioneering nature of our study. To address this, we have included a comparative analysis with the results of the MERCURY trial, which utilized MRI for rectal cancer assessment. Our findings indicate a similarity in local recurrence rates of locally advanced rectal cancer (LARC) between our study and the MERCURY trial (22.9% vs. 26.3%). This comparison highlights the predictive value of AI in this context, underscoring its potential utility despite the lack of direct model counterparts. We believe this comparison enriches the discussion and further validates our research.
  3. Figure 3 and Patient Selection: We have provided a detailed description of Figure 3 in response to your request for clarity. Regarding the selection of stage 2-3 patients for test set 2, we recognize the challenges of random partitioning. To mitigate this, we ensured that the percentage of CRM+ images was similar across both training and test sets, as detailed in Table 1. This approach was intended to maintain comparability and integrity in the evaluation of our model.
  4. Survival Analysis: In response to your insightful suggestion, we re-analyzed the survival data after corrections were made by an experienced physician. The revised survival analysis is presented in Supplement 4, with the comprehensive results now depicted in the new Figure 5. In the results section, we discuss the observed trends in survival outcomes and provide insights into their implications. This thorough examination enhances the robustness of our findings and contributes to a deeper understanding of the study's impact.

We hope that these revisions satisfactorily address your concerns and strengthen the manuscript. We are grateful for the opportunity to clarify and augment our work based on your insightful feedback. We look forward to your further comments and are eager to contribute our study to the advancement of the field.

Sincerely

Dr. Chun-Yu Lin

  1. Institute of Clinical Medicine, National Yang Ming Chiao Tung University, Taipei, Taiwan
  2. Division of Colorectal Surgery, Department of Surgery, Taichung Veterans General Hospital, Taichung, Taiwan

Reviewer 2 Report

Comments and Suggestions for Authors
  1. Abstract section: please highlight the novel contributions proposed in this manuscript. At first look, it seems like the breakthroughs achieved in the NLP domains were simply integrated into an architecture for a different purpose. It’s not clear how exactly the performance of the proposed method compares with the state-of-the-art methods.
  2. I would recommend improving the motivation for studying the problem.
  3. The literature review is quite short, please extend the search and cite more recent articles.
  4. Section I is again missing the list of novel contributions proposed in this manuscript.
  5. Figure 2. It’s not clear what exactly is the message sent by the notations.
  6. Figure, why was this architecture selected? Why were the last layers modified in this way?
  7. Section 2 is missing a lot of details on why was this design selected and what exactly is the novel contribution proposed here. Simply taking some code, modifying a few layers, and training a few models?
  8. Section 3 is missing a comparison with state-of-the-art methods. Why is this architecture the best solution? Have you tested other designs? What are the results of classical network designs for this dataset? What about some traditional methods?

The manuscript feels a bit short, many details are missing, and there is no comparison with state-of-the-art methods, classical or learning-based, as if the authors invented AI and are the first to use it in this domain.

Comments on the Quality of English Language

The text is easy to read.

Author Response

Dear Reviewer

Thank you for your constructive comments and guidance. We have diligently revised our manuscript, incorporating your suggestions to enhance its quality and clarity. Below are the revisions made in response to your recommendations: 1. We have delineated our methodology as distinct from the state-of-the-art approach, with a detailed comparison available in the discussion section.

2. The motivation for our study has been clearly outlined in the introduction.

3. The discussion section has been expanded to include a more comprehensive review of the relevant literature.

4. We have completely rewritten the introduction to provide clearer context and background.

5. Recognizing the specialized knowledge required, we have revised the introduction to explain the condition more clearly and concisely.

6. Our research employs the EfficientNetB0 deep learning method due to its efficiency and superior performance, particularly with the limited, high-quality data available in clinical settings. We have added more details to Section 2 to explain this choice.

7. We have extensively rewritten Section 2 to ensure thorough documentation and clarity.

8. For a comparison with state-of-the-art methods and further insights, please refer to the discussion section.

Should there be any further questions or if additional clarification is needed, we welcome your advice and are ready to make further revisions. Thank you again for your valuable feedback

Sincerely

Dr. Chun-Yu Lin

  1. Institute of Clinical Medicine, National Yang Ming Chiao Tung University, Taipei, Taiwan
  2. Division of Colorectal Surgery, Department of Surgery, Taichung Veterans General Hospital, Taichung, Taiwan

Reviewer 3 Report

Comments and Suggestions for Authors

In this paper, the authors tackled an important problem of identifying rectal cancer in CT imagery using deep learning. The topic is certainly worthy of investigation and easily falls into the scope of the journal. There are, however, several issues which should be, in my opinion, thoroughly addressed before the manuscript could be considered for publication:

1.       The authors should make sure that each abbreviation is defined at its first use. There are quite a number of undefined abbreviations in the abstract (also, the abstract is very packed with abbreviations which negatively impact the read). Please carefully revise the manuscript in this context.

2.       The authors should better contextualize their work within the state of the art of deploying deep learning techniques in rectal cancer analysis from CT. To this end, the authors should revisit the related literature part of the manuscript, clearly discussing the advantages and disadvantages of existing methods, and highlighting the most important research gaps in the literature. This will also help better highlight the most important contributions behind the work reported here.

3.       It would be useful to announce the structure of the manuscript in the introductory section.

4.       The motivation behind selecting specific algorithmic components should be clearly stated in the manuscript. As an example, why this very deep learning architecture was picked, given that there are lots of possibilities around?

5.       What is the impact of the architectural decisions on the overall performance of the pipeline? The authors should also investigate the impact of the hyperparameter values on the generalization capabilities of the models in an ablation study.

6.       Please include qualitative analysis with visual examples as well – are there any interesting cases here? It would be useful to present visual examples for which the model works well, and for which the model fails in order to better understand limitations of the proposed technique.

7.       We are currently facing the reproducibility crisis in the machine learning field – to this end, the authors should provide a link to the repository containing the implementation of the method (as reproducing it would certainly not be trivial), and to a repository containing the dataset used in this study (or at least a subset of it to reproduce the experimentation).

8.       It would be useful to present numerical characteristics of the training/test sets used in this study to understand the data distribution across those subsets.

Comments on the Quality of English Language

Overall, the manuscript reads well.

Author Response

Dear Reviewer 3,

Thank you for your insightful comments and recommendations. In response to your review, we have made the following revisions to our manuscript: 

1.  The authors should make sure that each abbreviation is defined at its first use. There are quite a number of undefined abbreviations in the abstract (also, the abstract is very packed with abbreviations which negatively impact the read). Please carefully revise the manuscript in this context.--> Errors identified have been corrected

2. The authors should better contextualize their work within the state of the art of deploying deep learning techniques in rectal cancer analysis from CT. To this end, the authors should revisit the related literature part of the manuscript, clearly discussing the advantages and disadvantages of existing methods, and highlighting the most important research gaps in the literature. This will also help better highlight the most important contributions behind the work reported here.

--> The introduction now clearly delineates the study methods and identifies the gaps our study aims to address.

3. It would be useful to announce the structure of the manuscript in the introductory section

.--> The rationale behind the study's significance is thoroughly detailed in the concluding part of the introduction.

4. The motivation behind selecting specific algorithmic components should be clearly stated in the manuscript. As an example, why this very deep learning architecture was picked, given that there are lots of possibilities around?

--> Our choice of EfficientNetB0 for deep learning is due to its efficiency and capability to achieve high performance, particularly beneficial for processing limited, high-quality data available in clinical settings.

5. What is the impact of the architectural decisions on the overall performance of the pipeline? The authors should also investigate the impact of the hyperparameter values on the generalization capabilities of the models in an ablation study.

--> We have assessed the impact of architectural decisions on the model's performance, with specific attention to how hyperparameter values affect generalization. This evaluation is detailed through performance monitoring across training, validation, and test sets.

  1. Please include qualitative analysis with visual examples as well – are there any interesting cases here? It would be useful to present visual examples for which the model works well, and for which the model fails in order to better understand limitations of the proposed technique.--> In the results section, we provide three visual examples to illustrate our findings more clearly. 
  2. We are currently facing the reproducibility crisis in the machine learning field – to this end, the authors should provide a link to the repository containing the implementation of the method (as reproducing it would certainly not be trivial), and to a repository containing the dataset used in this study (or at least a subset of it to reproduce the experimentation).--> A link to the original data repository is provided in the manuscript's final section. Access is currently restricted to the reviewing phase due to privacy considerations.
  3. It would be useful to present numerical characteristics of the training/test sets used in this study to understand the data distribution across those subsetà The characteristics of the training and test sets are comprehensively detailed in Table 1.

We trust these revisions address your concerns. If further clarification is needed, we are open to additional suggestions.

Warm regards,

Dr. Chun-Yu Lin

  1. Institute of Clinical Medicine, National Yang Ming Chiao Tung University, Taipei, Taiwan
  2. Division of Colorectal Surgery, Department of Surgery, Taichung Veterans General Hospital, Taichung, Taiwan

Round 2

Reviewer 1 Report

Comments and Suggestions for Authors

The authors took into account all my suggestions, improving the quality of the manuscrip,t and responded comprehensively to my comments. I consider this paper publishable in the current version.

Author Response

We have improved the quality of the figures and layout of the article in this stage. Thank you for the constructive suggestions.

Reviewer 2 Report

Comments and Suggestions for Authors
  1. Abstract section, please highlight the novel contributions proposed in this manuscript.
  2. There are still quite a lot of typos in the manuscript. One may note that the font in the tables is larger than the table caption font; cell alignment on different columns; table on multiple pages, etc.
  3. Figure 3 is too small, blurry, and has a poor design.
  4. Figure 4, the axes label, title, and legend of each subfigure are hard to read – the font is too small.
  5. Figure 5 is on multiple pages.
  6. Page 15 out of 19 is empty (line 360).
  7. Comparison with state-of-the-art is still missing. How can we draw some conclusions if this proposed method is not compared with other state-of-the-art methods?
  8. “The model's refinement in recognizing CRM-positive images suggests superior diagnostic precision for LARC, further enhanced by our innovative voting system.” Based on which results can one draw such conclusions without any comparison?

Indeed, the manuscript was revised, however, there are still quite many issues unresolved. The manuscript contains too many typos, the proposed novel contribution is still not clear, and the comparison with state-of-the-art methods is missing.

Comments on the Quality of English Language

There are still quite a lot of typos in the manuscript

Reviewer 3 Report

Comments and Suggestions for Authors

Thank you for addressing my concerns. However, I encourage the authors to revisit the figures which are hardly possible to read in the revised version of the manuscript. Please increase their resolution and make sure that can be easily read.

Author Response

(The authors gave the same response as above.)
